# Physicochemical Characteristics of Antimicrobials and Practical Recommendations for Intravenous Administration: A Systematic Review

**DOI:** 10.3390/antibiotics12081338

**Published:** 2023-08-19

**Authors:** Fabio Borgonovo, Massimiliano Quici, Antonio Gidaro, Davide Giustivi, Dario Cattaneo, Cristina Gervasoni, Maria Calloni, Elena Martini, Leyla La Cava, Spinello Antinori, Chiara Cogliati, Andrea Gori, Antonella Foschi

**Affiliations:** 1Department of Infectious Diseases, Luigi Sacco Hospital, ASST Fatebenefratelli-Sacco, University of Milan, 20157 Milan, Italy; 2Internal Medicine Unit, Luigi Sacco Hospital, ASST Fatebenefratelli-Sacco, Department of Biomedical and Clinical Sciences, University of Milan, 20157 Milan, Italy; 3Emergency Department and Vascular Access Team ASST Lodi, 26900 Lodi, Italy; 4Unit of Clinical Pharmacology, Luigi Sacco Hospital, ASST Fatebenefratelli-Sacco, University of Milan, 20157 Milan, Italy

**Keywords:** antimicrobials, antifungals, antiprotozoals, antivirals, antimycotics, antibiotics, intravenous administration, thrombosis, pH, osmolarity, vesicant, central catheter, peripheral catheter

## Abstract

Most antimicrobial drugs need an intravenous (IV) administration to achieve maximum efficacy against target pathogens. IV administration is related to complications, such as tissue infiltration and thrombo-phlebitis. This systematic review aims to provide practical recommendations about diluent, pH, osmolarity, dosage, infusion rate, vesicant properties, and phlebitis rate of the most commonly used antimicrobial drugs evaluated in randomized controlled studies (RCT) till 31 March 2023. The authors searched for available IV antimicrobial drugs in RCT in PUBMED EMBASE^®^, EBSCO^®^ CINAHL^®,^ and the Cochrane Controlled Clinical trials. Drugs’ chemical features were searched online, in drug data sheets, and in scientific papers, establishing that the drugs with a pH of <5 or >9, osmolarity >600 mOsm/L, high incidence of phlebitis reported in the literature, and vesicant drugs need the adoption of utmost caution during administration. We evaluated 931 papers; 232 studies were included. A total of 82 antimicrobials were identified. Regarding antibiotics, 37 reach the “caution” criterion, as well as seven antivirals, 10 antifungals, and three antiprotozoals. In this subgroup of antimicrobials, the correct vascular access device (VAD) selection is essential to avoid complications due to the administration through a peripheral vein. Knowing the physicochemical characteristics of antimicrobials is crucial to improve the patient’s safety significantly, thus avoiding administration errors and local side effects.

## 1. Introduction

It has been estimated that over 80% of hospitalized patients receive intravenous (IV) therapy [1]. This route of delivery has important advantages: it allows the administration of medications that cannot be orally absorbed, it permits reaching the highest drug’s bioavailability, and it can maintain a constant therapeutic dosage over time through continuous infusion. The most common drugs administered through vascular access devices (VADs) are fluids and nutrients, blood, or medications (especially antimicrobial agents and chemotherapy drugs) [2,3]. The catheter’s tip position differentiates these devices into two categories: central or peripheral (CVADs and PVADs. respectively). For CVADs, the catheter tip is located in the cavo-atrial junction or the lower third of the superior vena cava [3]. As regards the correct position of the tip for CVAD in the inferior vena cava, the literature does not provide clear recommendations. To date, two positions have been accepted, either near the right atrium or above the confluence of renal veins [4]. Devices with tips located in any other vein are considered PVADs. PVADs are represented by short peripheral catheters, long peripheral catheters, and midlines [5].

The appropriate type of VAD is selected according to the prescribed therapy, its duration, the patient’s characteristics, and the setting (inpatient or outpatient) [3]. Chemical characteristics of medications have a deep impact on VAD selection. CVAD tips are placed in large caliber veins guaranteeing the possibility of infusing any type of solution and drawing repeated blood samples. Furthermore, PVADs should not be utilized with low (<5) or high (>9) pH medications, high osmolarity drugs (>600 mOsm/L), and any other medication that can cause tissue damage in the event of leakage from the vascular lumen, such as vesicant drugs. Emphasizing the concept of a large caliber vein, the use of PVAD for the administration of drugs that are incompatible with a peripheral route is allowed in particular settings (such as the emergency department), provided it is for a limited period [6].

Peripheral veins are a low-flow system. Therefore, the administration of an irritant drug through a PVAD may damage the endothelial layer of the intima causing thrombosis, and may be associated with inflammatory processes involving the tunica media with the occurrence of edema, infiltration, and more serious damage up to the vein wall rupture. The events described above represent the pathological correlates of thrombophlebitis [7].

Moreover, using vesicant drugs, the potential dislocation of the catheter can lead to tissue damage of different entities (from mild effects to tissue necrosis) in case of escape to subcutaneous tissues. It is very important to note that contact between subcutaneous tissues and the drug does not occur only in the event of a massive leakage due to mechanical events (such as catheter dislocation) but also due to altered permeability of the venous walls due to an inflammatory process affecting them [8]. It, therefore, appears clear that the use of vesicant drugs is particularly delicate and worthy of attention. On this topic, several institutions have drafted lists of drugs that should be delivered via CVADs for their harmful effect. These lists include vesicant chemotherapy drugs, parenteral nutrition with high osmolarity, some antibiotics, some antivirals, vasoactive amines, and many other commonly used medications. These lists should be available in every hospital unit and used as a guide in VAD selection [6,9,10].

Of all drugs, antibiotics are among the most commonly used medication in the world [11]. In recent years, the emergence of antimicrobial-resistant bacteria has led to the development of new molecules or the reuse of old ones, and the majority of them need to be administered intravenously [12]. Data about preferred line administration (central or peripheral), dosage, pH, osmolarity, diluent, and infusion rate of antimicrobials are difficult to find in the literature and are often contrasting when available due to the lack of standardized protocols in this setting.

The incorrect choice of a VAD could increase the risk of complications due to “chemically induced” phenomena (such as phlebitis or thrombophlebitis and tissue lesions).

The present systematic review aims to collect practical recommendations about diluent, pH, osmolarity, dosage, infusion rate, vesicant drugs, and phlebitis rate and advise caution (red flag) of the most commonly used antimicrobial drugs evaluated in randomized controlled trials till 31 March 2023, to help the institutions in constructing antimicrobial drugs policies about safe IV administration.

## 2. Materials and Methods

This systematic review was performed following the preferred reporting items for systematic reviews and meta-analyses (PRISMA) [13]. This systematic review and meta-analysis was registered on PROSPERO (CRD42023434665).

We searched for available IV antimicrobial drugs in randomized controlled trials (RCT) in PUBMED^®^ EMBASE^®^, EBSCO^®^ CINAHL^®^, and the Cochrane Controlled Clinical trials.

Search strings were developed with the assistance of a medical librarian and MesH term browser [https://meshb.nlm.nih.gov/ (accessed on 31 March 2023)] and contained terms and synonyms for “antimicrobial agents” (Anti-Infective Agent; Anti-Microbial Agent; Anti-Microbial; Agents; Anti-infective Agents; Antimicrobial Agent; Antimicrobial Agents; Microbicide; Microbicides) or “antibiotics” (Anti-Bacterial Agent; Anti-Bacterial Compound; Anti-Bacterial Compounds; Anti-Mycobacterial Agent; Anti-Mycobacterial Agents; Antibacterial Agent; Antibacterial Agents; Antibiotic; Antibiotics; Antimycobacterial Agent; Antimycobacterial Agents; Bacteriocidal Agent; Bacteriocidal Agents; Bacteriocide; Bacteriocides) or “antivirals” (Antiviral; Antiviral Agent; Antiviral Drug; Antiviral Drugs; Antivirals) or “antimycotics” or “antifungals” or “antiprotozoals” and “new” and “intravenous”.

No MesH terms were found for “antimycotics”, “antifungals”, “antiprotozoals” “new”, and “intravenous”.

Only RCTs till 31 March 2023 were analyzed.

The exclusion criteria were:In vitro and/or animal studies,Papers not written in English;Papers not about antimicrobial agents;Papers about non-IV antimicrobial agents;Papers about antimicrobial drugs withdrawn due to adverse events or “out-of-market”;Papers about antimicrobial agents under investigation in phase II or III or waiting for approval.

All the papers that did not match the exclusion criteria were analyzed.

According to the literature, the authors have entered the warning of a “red flag” for solutions with a pH < 5 or >9, for solutions with osmolarity >600 mOsm/L, and for antimicrobial agents with a reported incidence of thrombophlebitis if infused in peripheral veins ≥5% or with vesicant properties [14].

Two authors (M.Q., F.B.) independently screened all papers for eligibility based on titles and abstracts through Covidence software. Relevant studies were then reviewed independently as full text for final inclusion. A third investigator (A.G.) resolved any inclusion disagreements. Finally, a hand search through the reference lists of included articles was conducted, and expert recommendations were screened for inclusion as well.

Considering that not all RCT reviewed reported the incidence of thrombophlebitis, after identifying the antimicrobial drugs, we searched for each drug’s chemical features in https://www.drugs.com/ (accessed on 31 March 2023), https://www.fda.gov/Drugs/ (accessed on 31 March 2023), drug data sheets and recently published scientific papers on the topic.

Three authors (F.B., A.F., and D.C.) reviewed the drug list, compared the various available sources, and resolved controversial data.

## 3. Results

A total of 931 papers were found, and 392 articles were removed before screening because they were not related to antimicrobial therapy. During the screening phase, 21 papers were excluded because they had been conducted on animals. A total of 216 articles treated non-IV antimicrobial agents. At last, we erased from our group 37 articles about antimicrobial agents under investigation in phases II and III or waiting for approval, and 33 papers about antimicrobial drugs were withdrawn due to adverse events or being “out-of-market”. A total of 232 papers were eventually analyzed in our systematic review (Figure 1).

A total of 82 antimicrobials (61 antibiotics, eight antivirals, 10 antifungals, and three antiprotozoals) were identified.

### 3.1. Antibiotics

Thirty-seven of the identified antibiotics meet the identified criteria to be considered at risk (high phlebitis reports (≥5%), very high/low pH, high osmolarity values, or vesicant properties). Among these, 14 have an extreme pH, 10 have a high report of phlebitis in the literature, and 12 have both characteristics. Notably, only Vancomycin has to be considered a vesicant drug. None have osmolarity values >600 mOsm/L, 24 can be infused through PVADs safely. Most of the antibiotics (64%) require dextrose 5% in water (D5W) or normal saline 0.9% NaCl (0.9% NaCl) as diluent. Six drugs need to be administered only with D5W, while saline infusion with 0.9% NaCl is essential for 14 of them. The infusion time is between a minimum of 15 min and a maximum of 3 h.

For further details on pH, osmolarity, dosage, infusion rate, and the “red flag” for each studied antimicrobial, see Table 1.

### 3.2. Antivirals

Seven antivirals meet the aforementioned precautionary criteria, one (Zanamivir) for low pH, two for high risk of phlebitis (Foscarnet both by induction and by maintenance), Ganciclovir and Oseltamivir for extreme pH and high risk of phlebitis, Remdesivir for extreme pH and vesicant characteristics, and Acyclovir for the presence of all the dangerous characteristics. Acyclovir and Foscarnet could be diluted either with D5W or 0.9% NaCl. The remaining antivirals need to be administered only with 0.9% NaCl. The infusion time is between a minimum of 30 min and a maximum of 2 h.

For further details on pH, osmolarity, dosage, infusion rate, and the “red flag” for each studied antimicrobial, see Table 2.

### 3.3. Antifungals

All antifungals reach the “red flag” criteria. Notably, only Amphotericin B has to be considered a vesicant drug. Five antifungals require D5W or 0.9% NaCl as diluent. Amphotericin B and Amphotericin B liposomal need to be administered only with D5W. Saline infusion with 0.9% NaCl is fundamental for Caspofungin, Fluconazole, and Posaconazole. The infusion time is between a minimum of 30 min and a maximum of 4 h.

For further details on pH, osmolarity, dosage, infusion rate, and the “red flag” about each studied antimicrobial, see Table 3.

### 3.4. Antiprotozoals

Eflornithine for the high rate of phlebitis described in the literature and Quinine for extreme pH need a cautious approach. The former must be administered with a 0.9% NaCl solution; Quinine can be infused with both 0.9% NaCl solution and D5W. Artesunate can be administered via PVADs diluted in a 12 mL trisodium phosphate (Na_3_PO_4_) solution through a 1–2-min infusion.

For further details on pH, osmolarity, dosage, infusion rate, and the “red flag” for each studied antimicrobial, see Table 4.

## 4. Discussion

Safe administration is a cornerstone of pharmacological treatment. VAD failure due to the development of complications (such as inflammatory events or leakage of the infusate from the venous wall) can lead to serious problems: tissue damage, delay in the administration of therapy, increased costs, prolongation of hospitalization or claims for compensation [242]. In 2022, the UK National Health Service Resolution published an analysis of extravasation-related litigation claims, which stated that the cost to the health system of compensation and treatment for extravasation amounted to £16 million over a ten-year observation period [243].

The use of peripheral catheters for the administration of intravenous therapy is the most used strategy worldwide. However, despite the undoubted advantages (cost containment, ease of use, and simplicity of management), this approach is weighed down by a considerable amount of complications [244].

For example, in a meta-analysis conducted in 2019, 35, studies were included (20.697 catheters used in 15.791 patients), and the incidence of phlebitis was 30.7 per 100 catheters (95% confidence interval: 27.2, 34.2). According to this study, longer dwelling time, antibiotics infusion, female gender, forearm insertion, infectious diseases, and the use of Teflon cannulae were the most important risk factors for phlebitis development [245]. Thus, a multifactorial approach is needed to address this issue. Several factors must be considered: patient conditions, number and quality of suitable veins, choice of vascular device, correct use of the device, and effective monitoring system. However, it is also important to consider the chemical characteristics of the drugs used. PH and osmolarity are well-recognized risk factors [246,247].

For these reasons, the authors decided to build up, through a systematic review, a list of the most commonly used antivirals, antifungals, antiprotozoals, and antimicrobial drugs commercialized or available soon. We collected data on diluent, pH, osmolarity, dosage, infusion rate, vesicant drugs, phlebitis rate, and presence/absence of “red flag” criteria for each studied medication in the hope that our efforts could help the institution to construct antimicrobial drugs policy about safe IV administration.

The results show that several antimicrobials have chemical characteristics that necessitate infusion through a CVAD due to their potentially dangerous effect on patients’ venous patrimony.

Surprisingly, only 39.3% of the antibiotics and 33.3% of the antiprotozoal have characteristics fully compatible with the use of a peripheral route, and this percentage decreases dramatically in the antiviral (12.5%) and antifungal (0%) groups.

Regarding the pH, high percentages of drugs with values <5 and >9 were found for each category, 60.7% for antibiotics, 87.5% for antivirals, and 70% and 33.3% for antifungals and antiprotozoals, respectively. It is important to remember that the pH values are scarcely modifiable by clinicians and that, as reported in the literature, the exposure of the vessel walls to pH values different from the blood’s pH, produces alterations of the endothelium with the consequent development of inflammatory phenomena and damage of the vessel wall integrity.

It is interesting to note that the majority of antimicrobials reported (75%) should be administered with rapid infusion times (below 60 min). It is, therefore, possible to speculate that even local contact is minimized due to the rapid transit of the solution (a phenomenon known as hemodilution) [248]. On the other hand, the endothelial damage caused by pH values could be exacerbated by the infusion rate, which, as emerges in the literature, is responsible for a certain amount of venous wall stress [249]. Furthermore, it is important to underline that if hemodilution could be a protective factor, it is also true that this is scarcely effective in low-flow systems (such as the superficial veins of the forearm) [250].

As regards osmolarity, the threshold beyond which a drug is not considered suitable for PVADs remains unclear [3]. However, administration of hypertonic IV solutions in a low-flow system, as peripheral veins are, is associated with osmotic changes that may damage the endothelial layer of the intima resulting in several adverse events such as “phlebitis”, “phlebothrombosis”, or “thrombophlebitis” [7,251]. Moreover, none of the antimicrobials examined have osmolarity >600 MOsm/L.

The overall percentage of antimicrobials requiring utmost caution is almost 70%, and in the “real world”, the choice to use a CVAD for administration should be the clear consequence.

Nevertheless, the authors are aware that the choice between PVADs and CVADs is often not simple in daily practice. It is essential to consider various elements: treatment duration, the conditions of the patient’s venous patrimony, and drug characteristics. In daily clinical practice, however, the immediate availability and placement of a CVAD is not always guaranteed, especially in small hospitals. Drugs that should preferably be delivered via a CVAD may be administered via a peripheral catheter pending the placement of a central access, especially in emergencies or to avoid delays in the start of proper antibiotic therapy.

Fortunately, in the last twenty years, the development and spread of peripherally inserted central catheters (PICCs) have revolutionized and redefined the world of CVAD, thanks to the low-risk insertion maneuver, thus avoiding life-threatening complications and the increase of health care professionals (both nurses and physicians) trained to insert.

In this changing scenario, an increase in the use of CVAD (especially PICCs) must lead to the necessary structural changes in healthcare facilities to ensure the effective and safe use of these devices (use of specific and standardized procedures and specifically trained personnel).

Therefore, the authors agree with the ERPIUP consensus regarding the correct indication of a peripheral catheter [6] and hope that the development of specific vascular access teams can also be effective in choosing the most appropriate device to use, as widely demonstrated in the most recent literature [252,253,254,255].

Although the physicochemical characteristics of chemotherapeutic drugs are well described in the literature, data are lacking for most antimicrobials, and it is very difficult to retrieve information on pH, osmolarity, required diluent, and infusion rate of each antimicrobial. Even when available, these data are often conflicting, comparing different documents. In the authors’ opinion, this constitutes a limitation of this review. However, this weakness underlines the need to standardize this information to obtain homogeneity in future guidelines.

Another limitation derives from the lack of pathophysiological description of phlebitis reported, making it impossible to discriminate between events of chemical origin from infective phlebitis or mechanical phlebitis. Like the previous limit, this also highlights the heterogeneity of the data available in the literature and the need for future developments on this issue.

Finally, we have to point out that commercially available antimicrobials are often realized by various pharmaceutical producers with the addition of different adjuvants or with structural modifications that can alter the biochemical characteristics of the administered drug, thus leading to minor or major vesicant properties and the ability to cause phlebitis. Therefore, the physicochemical characteristics of an original antimicrobial are usually different from that of the commercially available drug. This factor represents a further limitation of our study.

## 5. Conclusions

IV therapy is essential in everyday clinical practice. Although IV therapy is widely used, it is also known to be related to several complications. For this reason, the physicochemical properties of the delivered medication play a central role in VAD-related complications, especially when a vesicant or an irritant solution is delivered through the wrong vascular device. Knowledge of the physicochemical characteristics of the drug is fundamental in the selection of a suitable VAD to reduce the risk of vascular damage. All limitations considered, the list we created, in addition to the already available lists of other IV drugs, could be a useful tool to guide clinicians and institutions to improve the selection of the most appropriate VAD, thus avoiding administration errors or the onset of local side effects such as phlebitis and thrombosis.

## Figures and Tables

**Figure 1 antibiotics-12-01338-f001:**
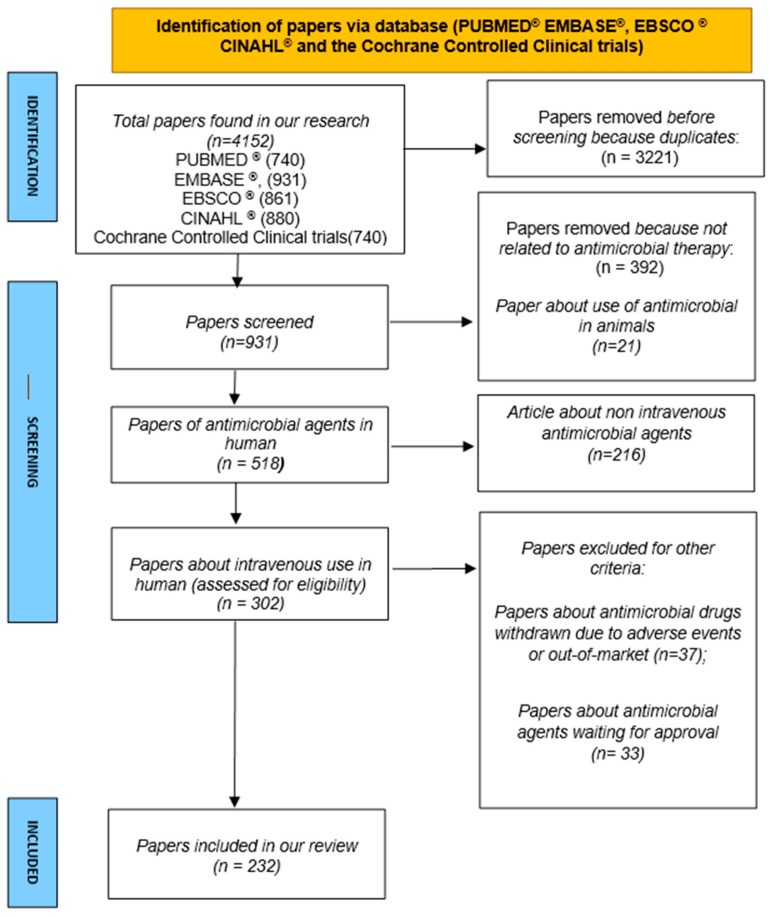
PRISMA flow diagram of our systematic review.

**Table 1 antibiotics-12-01338-t001:** List of antibiotics identified through our systematic review, with their physicochemical characteristics and practical recommendations for intravenous administration. (Phlebitis: Yes = incidence ≥ 5%; No = incidence < 5%; NA: not available; P: preferred; 0.9% NaCl: sodium chloride 0.9%; D5W: dextrose 5% in water). Red flag and vesicant properties: Red: central line preferred; Green: peripheral line preferred.

Antibiotics							
Name	Red Flag and Vesicant Properties	pH	mOsm/L	Phlebitis (Incidence ≥ 5%)	Diluent	Dosage	Infusion Rate
**Amikacin** [15,16,17,18,19]		3.5–5.5	312–383	No	0.9% NaCl/D5W	15 mg/kg	Over 30–60 min
**Amoxicillin/clavulanate** [20,21]		8.9	290–442	No	0.9% NaCl	1–2 g q 8 h	At least 30 min
**Ampicillin** [22,23]		8–10	238–372	Yes	0.9% NaCl	2 g q 4 h	Over 15 min
**Ampicillin/Sulbactam** [24,25]		8–10	320–400	Yes	0.9% NaCl	3 g q 6 h	Over 30 min
**Azithromycin** [26,27,28,29,30]		6.4–6.8	280–321	Yes	0.9% NaCl/D5W	500 mg q 24 h	At least 60 min
**Aztreonam** [31,32,33,34]		6	315–352	No	0.9% NaCl/D5W	2 g q 6–8 h	Over 20–60 min
**Cefazolin** [23]		6	315–352	No	0.9% NaCl	1–2 g q 8 h	Over 30 min
**Cefepime** [35,36,37,38,39]		4–6	362–396	No	0.9% NaCl/D5W	1–2 g q 8–12 h	Over 30 min
**Cefiderocol** [40,41]		5.2–5.8	NA	Yes	0.9% NaCl	2 g q 6–8 h	At least 180 min
**Cefmetazole** [42]		5.5–7.5	300	No	0.9% NaCl	1–2 g q 12 h	Over 30–60 min
**Cefodizime** [43,44]		4.5–6.5	280–320	No	0.9% NaCl/D5W	1–3 g q 8–12 h	Over 30–60 min
**Cefoperazone** [45]		6–8	265–335	No	0.9% NaCl/D5W	2–4 g/day divided q 12 h	Over 30–60 min
**Cefotaxime** [23,46,47,48,49]		5.2–5.4	339–411	No	0.9% NaCl/D5W	2 g q 6–8 h	At least 15 min
**Cefotetan** [24,50,51]		4–6.5	300	No	0.9% NaCl/D5W	1–2 g q 12 h	At least 20 min
**Cefoxitin** [23,42,50,51,52]		4–7.5	240–360	No	0.9% NaCl/D5W	2 g q 6–8 h	Over 30–60 min
**Ceftaroline** [31,32,53,54,55,56,57,58,59,60,61,62]		4.8–6.5	265	Yes	0.9% NaCl/D5W	600 mg q 8–12 h	At least 30–60 min
**Ceftazidime/Avibactam** [17,18,36,63,64,65,66,67,68,69,70,71,72,73]		6.7	335–386	No	0.9% NaCl/D5W	2.5 g q 8 h	At least 120 min
**Ceftizoxime** [46]		5.5–7.5	300–360	No	0.9% NaCl/D5W	1–2 g q 8–12 h	Over 30–60 min
**Ceftriaxone** [16,22,29,54,74,75,76,77,78]		6.6–6.7	423	No	0.9% NaCl	2 g q 12–24 h	At least 30 min
**Ceftolozane/Tazobactam** [79]		5.9–6.1	307–520	No	0.9% NaCl/D5W	1.5 g q 8 h	At least 60 min
**Cefuroxime** [30]		5.2–8.5	270–357	No	0.9% NaCl/D5W	0.75–1.5 g q 8 h	Over 15–60 min
**Ciprofloxacin** [49,65,66,67,80,81,82,83,84,85,86,87,88,89]		3.3–4.6	285	Yes	0.9% NaCl/D5W	400 mg q 8–12 h	Over 60 min
**Clarithromycin** [29,90,91]		5.2–5.4	275–301	No	0.9% NaCl/D5W	500 mg q 12 h	At least 60 min
**Chloramphenicol** [22]		7–8.5	600	No	0.9% NaCl/D5W	50–100 mg/kg/day	Over 30–60 min
**Trimethoprim/sulfamethoxazole** [92]		8.6	316–417	Yes	0.9% NaCl/D5W	5–20 mg TMP/kg q 6–12 h	Over 60–90 min
**Dalbavancin** [93,94,95]		4.5	246	No	D5W	1.5 g single dose	At least 30 min
**Daptomycin** [96,97]		4.7–6.8	263–364	No	0.9% NaCl	4–12 mg/kg q 24 h	Over 30 min
**Delafloxacin** [98,99,100,101]		NA	NA	Yes	0.9% NaCl/D5W	300 mg q 12 h	Over 60 min
**Doripenem** [102,103,104,105]		4.5–5.5	NA	Yes	0.9% NaCl/D5W	500 mg q 8 h	Over 60 min
**Doxycycline** [78,106]		1.8–3.3	292–310	No	0.9% NaCl/D5W	100 mg q 12 h	At least 60 min
**Eravacycline** [107]		NA	NA	Yes	0.9% NaCl	1 mg/kg q 12 h	At least 60 min
**Erythromycin** [29,30,108,109]		6.5–7.7	266–305	Yes	0.9% NaCl	15–20 mg/kg/day	At least 60 min
**Ertapenem** [77,110]		7.5	293–395	No	0.9% NaCl	1 g q 24 h	At least 30 min
**Fosfomycin** [96,111]		7.7	501–579	No	D5W	12–24 g/day	At least 60 min
**Flucloxacilllin** [112]		4.5–6.5	NA	No	0.9% NaCl/D5W	1–2 g q 4–6 h	At least 20 min
**Garenoxacin** [97]		NA	NA	Yes	D5W	400–600 mg/day	At least 60 min
**Gatifloxacin** [76,113,114]		3.5–5.5	NA	Yes	D5W	400 mg q 24 h	At least 60 min
**Gentamicin** [69,82,83,84,85,115,116,117]		3–5.5	280–318	No	0.9% NaCl/D5W	5 mg/kg/day	Over 30 min
**Imipenem/cil/rel** [37,40,80,105,118,119,120,121,122]		6.5–7.5	310–396	Yes	D5W	1.25 q 6 h	At least 30 min
**Lefamulin** [123,124]		4.5–5.5	280–340	Yes	0.9% NaCl	150 mg q 12 h	At least 60 min
**Levofloxacin** [63,89,103,125,126,127,128,129,21]		3.8–5.8	250–323	Yes	0.9% NaCl/D5W	750 mg q 24 h	Over 60 min
**Linezolid** [33,101,130,131,132,133,134,135,136,137,138,139]		4.8	283–309	No	0.9% NaCl/D5W	600 mg q 12 h	At least 30–120 min
**Meropenem** [86,107,111,120,121,140,141]		7.8–7.9	301–451	No	0.9% NaCl/D5W	1–2 g q 8 h	At least 30 min
**Meropenem/Vaborbactam** [142]		NA	NA	Yes	0.9% NaCl/D5W	2 g q 8 h	At least 180 min
**Metronidazole** [37,72,80,88]		5–7	310	No	0.9% NaCl/D5W	15 mg/kg then 7.5 mg/kg q 6 h	Over 60 min
**Moxifloxacin** [20,143,144,145]		4.1–4.6	270–320	No	0.9% NaCl/D5W	400 mg q 24 h	Over 60 min
**Netilmicin** [146]		6.5–7.5	274–306	No	0.9% NaCl/D5W	1.7–2 mg/kg/day	At least 120 min
**Ofloxacin** [16]		3.8–5.8	252	Yes	0.9% NaCl/D5W	200–400 mg q 12 h	At least 30 min
**Oritavancin** [147]		3.6–3.8	300	Yes	D5W	1.2 g single dose	Over 60–180 min
**Omadacyclin** [148]		4.2	259–279	Yes	0.9% NaCl/D5W	100–300 mg q 12 h	Over 30–60 min
**Oxacillin** [112]		6–8.5	270–398	Yes	0.9% NaCl/D5W	2 g q 4 h	Over 30–60 min
**Piperacilline/tazobactam** [35,81,88,104,122,140,141,149]		6.2–6.7	270–355	No	0.9% NaCl/D5W	4.5 g q 6–8 h	At least 30 min
**Plazomicin** [125,150]		6.5	NA	No	0.9% NaCl	15 mg/kg q 24 h	At least 30 min
**Rifampicin** [151]		7.8–8.8	264–300	No	0.9% NaCl/D5W	10 mg/kg/day	At least 180 min
**Teicoplanin** [152,153,154,155]		7.5–7.7	282–304	No	0.9% NaCl/D5W	6–12 mg/kg q 12–24 h	At least 30 min
**Telavancin** [156]		4.5	300	No	0.9% NaCl/D5W	10 mg/kg/day	At least 60 min
**Tedizolid** [131,157]		7.4–8.1	298	No	0.9% NaCl	200 mg q 24 h	Over 60 min
**Ticarcillin/clavulanic acid** [116,21]		6–8	573	Yes	0.9% NaCl	3.2 g q 8 h	At least 30 min
**Tigecycline** [45,101,158,159,160]		5–5.4	259–296	No	0.9% NaCl/D5W	100 mg/day	Over 30–60 min
**Tobramycin** [142]		3–6.5	290–316	No	0.9% NaCl/D5W	5.1–7 mg/kg/day	Over 30–60 min
**Vancomycin** [31,55,112,133,139,139,161,162,163]	Vesicant drug	2.5–4.5	235–300	Yes	0.9% NaCl/D5W	15–20 mg/kg q 8–12 h	At least 60 min

**Table 2 antibiotics-12-01338-t002:** List of antivirals identified through our systematic review, with their physicochemical characteristics and practical recommendations for intravenous administration. (Phlebitis: Yes = incidence ≥ 5%; No = incidence < 5%; NA: not available; P: preferred; 0.9% NaCl: sodium chloride 0.9%; D5W: dextrose 5% in water). Red flag and vesicant properties: Red: central line preferred.

Antibiotics							
Name	Red Flag and Vesicant Properties	pH	mOsm/L	Phlebitis (Incidence ≥ 5%)	Diluent	Dosage	Infusion Rate
**Acyclovir** [164,165,166,167,168,169,170,171,172,173,174,175,176]	Vesicant drug	10.5–11.6	285–323	Yes	0.9% NaCl/D5W	5–12.5 mg/kg q 8 h	At least 60 min
**Cidofovir** [177,178]		7.4–8.5	290	No	0.9% NaCl	5 mg/kg once weekly × 2 weeks, then once every other week	Over 60 min
**Foscarnet (induction)** [179,180,181]		7.4	271–315	Yes	0.9% NaCl/D5W	90 mg/kg q 12 h	Over 90–120 min
**Foscarnet (mainteinance)** [179,180,181]		7.4	271–315	Yes	0.9% NaCl/D5W	90–120 mg/kg q 24 h	Over 120 min
**Ganciclovir** [182,183]		11	290–320	Yes	0.9% NaCl	5 mg/kg q 12 h	At least 60 min
**Oseltamivir** [184,185]		4	NA	Yes	0.9% NaCl	100 mg q 12 h	At least 120 min
**Remdesivir** [186,187,188,189,190,191,192,193,194,195,196,197,198]	Vesicant drug	3	NA	NA	0.9% NaCl	200–100 mg/day	At least 30–120 min
**Zanamivir** [185,199]		4.5–6.5	NA	No	0.9% NaCl	600 mg q 12 h	Over 30 min

**Table 3 antibiotics-12-01338-t003:** List of antifungals identified through our systematic review, with their physicochemical characteristics and practical recommendations for intravenous administration. (Phlebitis: Yes = incidence ≥ 5%; No = incidence < 5%; NA: not available; P: preferred; 0.9% NaCl: sodium chloride 0.9%; D5W: dextrose 5% in water). Red flag and vesicant properties: Red: central line preferred.

Antifungals							
Name	Red Flag and Vesicant Properties	pH	mOsm/L	Phlebitis (Incidence ≥ 5%)	Diluent	Dosage	Infusion Rate
**Amphotericin B** [200,201,202,203,204,205,206,207,208,209,210,211,212,213]	Vesicant drug	5.7	256	Yes	D5W	0.3–1 mg/kg/day	At least 240 min
**AmphotericinB (liposomal)** [200,211,212,213]	Vesicant drug	5–6	280–309	Yes	D5W	3–5 mg/kg/day	At least 120 min
**Anidulafungin** [214,215]		3.5–5.5	209–269	Yes	0.9% NaCl/D5W	200–100 mg q 24 h	Max speed: 1.1 mg/min
**Caspofungin** [208]		6.6	256–298	Yes	0.9% NaCl	70–50 mg q 24 h	At least 60 min
**Fluconazole** [203,204,206,209,215,216,217,218]		4–8	299–316	No	0.9% NaCl	100–800 mg/day	Over 60–120 min
**Isavuconazole** [219,220,221]		1.9–2.6	NA	Yes	0.9% NaCl/D5W	372 mg q 8 h–24 h	Over 60 min
**Itraconazole** [222,223]		4.8	315	Yes	0.9% NaCl/D5W	200 mg q 12 h–24 h	At least 30 min
**Micafungin** [205,208,217,224]		4.3–5.9	281–307	Yes	0.9% NaCl/D5W	100–150 mg q 24 h	At least 60 min
**Posaconazole** [225,226,227,228]		2.6	281–292	Yes	0.9% NaCl	300 mg q 12 h–q 24 h	At least 30 min
**Voriconazole** [214,229]		4.4–5.5	191–242	No	0.9% NaCl/D5W	6–4 mg/kg q 12 h	Over 60–180 min

**Table 4 antibiotics-12-01338-t004:** Our systematic review identifies antiprotozoals with their physicochemical characteristics and practical recommendations for intravenous administration. (Phlebitis: Yes = incidence ≥ 5%; No = incidence < 5%; NA: not available; P: preferred; 0.9% NaCl: sodium chloride 0.9%; D5W: dextrose 5% in water, Na_3_PO_4_: trisodium phosphate). Red flag and vesicant properties: Red: central line preferred; Green: peripheral line preferred.

Antiprotozoals							
Name	Red Flag and Vesicant Properties	pH	mOsm/L	Phlebitis (Incidence ≥ 5%)	Diluent	Dosage	Infusion Rate
**Artesunate** [230]		7.2–7.7	305–317	No	12 mL Na_3_PO_4_	2–4 mg/kg/day	1–2 min
**Eflornithine** [231,232]		NA	NA	Yes	0.9% NaCl	100 mg/kg q 6 h	60 min
**Quinine** [233,234,235,236,237,238,239,240,241]		1.5–3	NA	NA	0.9% NaCl/D5W	20 mg/kg then 10 mg/kg q 8 h	Over 240 min

## Data Availability

Data is contained within the article. The data presented in this study are available in the present article.

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
