# Peer review of "Physicochemical Characteristics of Antimicrobials and Practical Recommendations for Intravenous Administration: A Systematic Review"

_antibiotics, 2023, doi:10.3390/antibiotics12081338_

Round 1

Reviewer 1 Report

The authors profiled the physicochemical characteristics of some antimicrobials from drug data sheets, and scientific papers to avoid administration errors and local side effects. It’s good attempt to supply the detailed information for the drugs administration. However, we have to make clear that commercially available antimicrobial drugs have been well deceived with addition of adjuvants or structural modification to cover most of these concerns from the authors. The physicochemical characteristics of original antimicrobials are less instructive for the administration of the drugs. In addition, the data quality from the paper is not enough to conclude the effects and safety of antimicrobials. To achieve a reliable review to sever for the drug administration, the data from clinical trial reports should be included.

The manuscript was well organized, and there were few grammar problems. The language was fluent with good readability.

Author Response

Dear Editor,

Please find enclosed our revised version of the manuscript entitled “Physicochemical characteristics of antimicrobials and practical recommendations for intravenous administration: a Systematic Review”. We would like to thank the Editor and the reviewers for their constructive criticisms that for sure substantially improved our paper. We hope that the paper will be suitable for publication. We have made a point by point answer of referees’ comments below.

Answers to Reviewer 1

Reviewer 1: The authors profiled the physicochemical characteristics of some antimicrobials from drug data sheets, and scientific papers to avoid administration errors and local side effects. It’s good attempt to supply the detailed information for the drugs administration. However, we have to make clear that commercially available antimicrobial drugs have been well deceived with addition of adjuvants or structural modification to cover most of these concerns from the authors. The physicochemical characteristics of original antimicrobials are less instructive for the administration of the drugs.

Response to Reviewer 1 comment: We thank the reviewer for his/her comment and criticisms. We agree with reviewer opinion. This limitation was added in the discussion paragraph. Below you can find the new sentence:

“Finally, we have to point out that commercially available antimicrobials are often realized by various pharmaceutical producers with the addition of different adjuvants or with structural modifications that can alter the biochemical characteristics of the administered drug, thus leading to minor or major vesicant properties and ability to cause phlebitis. Therefore, the physicochemical characteristics of an original antimicrobial are usually different from that of the commercially available drug. This factor represents a further limitation of our study.”

Reviewer 1: In addition, the data quality from the paper is not enough to conclude the effects and safety of antimicrobials. To achieve a reliable review to sever for the drug administration, the data from clinical trial reports should be included

Response to Reviewer 1 comment: We thank the reviewer for his/her comment and criticisms. The data from clinical trial reports have just been included in the column “phlebitis” in which we decided to define as safe all the drugs with a reported incidence of phlebitis less than 5% in the RCT examined. This was declared in page 3 line 131. In order to improve the paper and clarify this concept, the table and the text were changed.

Reviewer 1: The manuscript was well organized, and there were few grammar problems. The language was fluent with good readability.

Response to Reviewer 1 comment: We thank the reviewer for his/her comment and suggestion.

English language has been reviewed.

Answers to Reviewer 2

Reviewer 2: The aim of the study was to collect practical recommendations about diluent, pH, osmolarity, dosage, infusion rate, vesicant drugs, phlebitis rate, and advise to caution of the most commonly used antimicrobial drugs evaluated in randomized controlled trials till 31st March 2023, to help the institution in construct antimicrobial drugs policy about safety intravenous administration.

In general, the structure and concise working of the manuscript are adequate, which greatly facilitates its understanding by potential readers. On the other hand, tables and figures in the manuscript are adequate.

Response to Reviewer 2 comment:  We thank the reviewer for his/her positive comment.

Reviewer 2: Comment 1: In my opinion, the search equation can be improved, have the authors considered including the term MesH, Anti-Bacterial Agents? Likewise, other terms, “antibiotics” and “Bacteriocides” must be included to consult the databases using the title and the abstract field (Title/Abstract). It is important to include all terms to minimize the number of papers not recovered by the review.

Response to Reviewer 2 comment:  We thank the reviewer for his/her suggestion. Now we have specified in the “materials and methods” section the use of MesH term browser: (https://meshb.nlm.nih.gov/ )  to expand our research.

Antimicrobial agents MesH term added are: Anti-Infective Agent; Anti-Microbial Agent; Anti-Microbial; Agents; Antiinfective Agents; Antimicrobial Agent; Antimicrobial Agents; Microbicide; Microbicides

Antibiotics MesH term added are: Anti-Bacterial Agent; Anti-Bacterial Compound; Anti-Bacterial Compounds; Anti-Mycobacterial Agent; Anti-Mycobacterial Agents; Antibacterial Agent; Antibacterial Agents; Antibiotic; Antibiotics; Antimycobacterial Agent; Antimycobacterial Agents; Bacteriocidal Agent; Bacteriocidal Agents; Bacteriocide; Bacteriocides

Antivirals MesH term added are: Antiviral; Antiviral Agent; Antiviral Drug; Antiviral Drugs; Antivirals

No MesH terms were found for “antimycotics”, “antifungals”, “antiprotozoals” “new” and “intravenous”

The number of paper found was the same reported in the first version of the paper.

We have specified in the new search equation all the MesH term used. Below you can find the new sentence:

“Search strings were developed with the assistance of a medical librarian and MesH term browser261 and contained terms and synonyms for “antimicrobial agents”(Anti-Infective Agent; Anti-Microbial Agent; Anti-Microbial; Agents; Antiinfective Agents; Antimicrobial Agent; Antimicrobial Agents; Microbicide; Microbicides) OR “antibiotics” (Anti-Bacterial Agent; Anti-Bacterial Compound; Anti-Bacterial Compounds; Anti-Mycobacterial Agent; Anti-Mycobacterial Agents; Antibacterial Agent; Antibacterial Agents; Antibiotic; Antibiotics; Antimycobacterial Agent; Antimycobacterial Agents; Bacteriocidal Agent; Bacteriocidal Agents; Bacteriocide; Bacteriocides) OR “antivirals” (Antiviral; Antiviral Agent; Antiviral Drug; Antiviral Drugs; Antivirals) OR “antimycotics” OR “antifungals” OR “antiprotozoals” AND “new” AND “intravenous”.

No MesH terms were found for “antimycotics”, “antifungals”, “antiprotozoals” “new” and “intravenous”.”

Reviewer 2: Comment 2: In the search equation, have the authors considered to include a search filter based on publication date instead of the term "new"?

Response to Reviewer 2 comment:  We thank the reviewer for his/her question. If we use a search filter based on publication date instead of the term “new” we will lose many drugs. For example, if we apply publication date after 2000 we lose:

Tobramicin (quote 1996) Teicoplanin (all quotes before 2000) Doxycycline (quote 1997, 1989); Cefazolin (quote 1982) and many other drugs.

Reviewer 2: Comment 3: In material and methods, the inclusion criteria of the articles are not defined?

Response to Reviewer 2 comment:  We thank the reviewer for his/her question. We decided to define only exclusion criteria, all the papers that didn’t match the exclusion criteria were analyzed. Below you can find the new sentence included in the methods section:

“All the papers that didn’t match the exclusion criteria were analyzed.”

Reviewer 2: Comment 4: Could the authors include the search equations of each database in the manuscript? I believe that with those search terms more than 740 Pubmed articles are retrieved.

Response to Reviewer 2 comment:  We thank the reviewer for his/her question. Below you can find the new search equation for Pubmed, improved with all the MesH terms suggested by the reviewer, that found 735 papers:

https://pubmed.ncbi.nlm.nih.gov/?term=%28%28%28%28%28%28%28%28%28%28%28Antivirals%29+OR+%28Antiviral+Drugs%29%29+OR+%28Antiviral+Drug%29%29+OR+%28Antiviral+Agent%29%29+OR+%28Antiviral%29%29+OR+%28%28%28%28%28%28%28%28%28%28%28%28%28%28%28Bacteriocides%29+OR+%28Bacteriocide%29%29+OR+%28Bacteriocidal+Agents%29%29+OR+%28Bacteriocidal+Agent%29%29+OR+%28Antimycobacterial+Agents%29%29+OR+%28Antimycobacterial+Agent%29%29+OR+%28Antibiotics%29%29+OR+%28Antibiotic%29%29+OR+%28Antibacterial+Agents%29%29+OR+%28Antibacterial+Agent%29%29+OR+%28Anti-Mycobacterial+Agents%29%29+OR+%28Anti-Mycobacterial+Agent%29%29+OR+%28Anti-Bacterial+Compounds%29%29+OR+%28Anti-Bacterial+Compound%29%29+OR+%28Anti-Bacterial+Agent%29%29%29+OR+%28%28%28%28%28%28%28%28%28Microbicides%29+OR+%28Microbicide%29%29+OR+%28Antimicrobial+Agents%29%29+OR+%28Antimicrobial+Agent%29%29+OR+%28Antiinfective+Agents%29%29+OR+%28Anti-Microbial+Agents%29%29+OR+%28Anti-Microbial+Agent%29%29+OR+%28Anti-Infective+Agent%29%29+OR+%28antimicrobial+agents%29%29%29+OR+%28antimycotics%29%29+OR+%28antifungals%29%29+OR+%28antiprotozoals%29%29+AND+%28new%29%29+AND+%28intravenous%29&filter=pubt.randomizedcontrolledtrial&sort=date

Reviewer 2 Report

The aim of the study was to collect practical recommendations about diluent, pH, osmolarity, dosage, infusion rate, vesicant drugs, phlebitis rate, and advise to caution of the most commonly used antimicrobial drugs evaluated in randomized controlled trials till 31st March 2023, to help the institution in construct antimicrobial drugs policy about safety intravenous administration.

In general, the structure and concise working of the manuscript are adequate, which greatly facilitates its understanding by potential readers. On the other hand, tables and figures in the manuscript are adequate.

Comment 1: In my opinion, the search equation can be improved, have the authors considered including the term MesH, Anti-Bacterial Agents? Likewise, other terms, “antibiotics” and “Bacteriocides” must be included to consult the databases using the title and the abstract field (Title/Abstract). It is important to include all terms to minimize the number of papers not recovered by the review.

 Comment 2: In the search equation, have the authors considered to include a search filter based on publication date instead of the term "new"?

 Comment 3: In material and methods, the inclusion criteria of the articles are not defined?

 Comment 4: Could the authors include the search equations of each database in the manuscript? I believe that with those search terms more than 740 Pubmed articles are retrieved.

Author Response

(The authors gave the same response as above.)

Round 2

Reviewer 1 Report

Most of my concerns have been addressed, and I would like to recommend the publication of this manuscript in our Journal.

Reviewer 2 Report

Manuscript has been improved